# Elucidation of Ischemic Mechanisms of Early Pressure Injury during Post-Decompression and Detecting Methods

**DOI:** 10.3390/diagnostics12092198

**Published:** 2022-09-09

**Authors:** Lu Chen, En Takashi, Ping Hou, Akio Kamijo, Daiji Miura, Jianglin Fan

**Affiliations:** 1Division of Basic & Clinical Medicine, Faculty of Nursing, Nagano College of Nursing, Komagane 399-4117, Japan; 2Department of Molecular Pathology, Faculty of Medicine, Graduate School of Medical Sciences, University of Yamanashi, Kofu 409-3898, Japan; 3School of Nursing, Yangzhou University, Yangzhou 225000, China

**Keywords:** ischemia, early PI, post-decompression, rat models, CRTT

## Abstract

Background: Pressure injuries (PIs) generally result from prolonged ischemia through localized skin compression, and ischemia persists and exacerbates damage even post-decompression. The mechanisms of ischemia post-decompression are still unclear, and appropriate methods for detection are lacking. Methods: We used blanchable erythema (BE) and early PI rat models. We assessed the perfusion using Evans Blue (EB) and thrombus formation under a light microscope. Furthermore, we performed a capillary refill time test (CRTT) to detect ischemia after depression coupled with the transparent disk method using a spectrophotometer. Results: Compared with the BE group, the early PI group showed significantly slow and insufficient perfusion, as determined by EB staining (*p* < 0.001). Histological observations revealed that ischemia during post-decompression of early PI was caused by a greater amount of thrombi. The CRTT results showed that although both groups exhibited varying degrees of insufficient refilling volume, the early PI group had significantly slower refilling than the BE group (*p* < 0.001), which persisted during the deterioration or disappearance of erythema. Conclusions: Our results showed that persistent ischemia caused by thrombi is an important cause of early PI deterioration post-decompression. Therefore, the performance of CRTT coupled with the transparent disc method may become a promising method for detecting ischemia post-decompression.

## 1. Introduction

Pressure injuries (PIs) are sores (ulcers) that occur on areas of the skin that are under pressure. PIs can be a serious problem in frail older adults in aging societies such as Japan. It is well-known that ischemia is essential for the occurrence of PIs [1]. However, ischemia can persist even after the reduction in the local skin pressure, although the mechanisms before and post-decompression are different. Moreover, post-decompression ischemia may also exacerbate PIs. Reichel et al. [2] found that multiple thrombi in small vessels accompanied by subsequent necrosis are consistent pathological findings in PIs and that the prolonged ischemia caused by the thrombi also produces necrosis [3]. At the same time, Gefen et al. [4] highlighted that ischemia exacerbated by inflammation, thrombosis, and reperfusion injury may lead to further tissue damage. This notion was supported by animal and clinical observations in which the administration of a single dose of ACTH to dissolve thrombosis was found to reduce the incidence of PIs [5,6].

Despite this, the mechanisms by which ischemia occurs post-decompression are not fully understood, and the optical method for the assessment of ischemia in routine clinical settings remains to be elucidated. Although MRI can be used to detect ischemia following PI post-decompression [7], the cost of such large instruments limits their practical use in clinical examinations. The lack of appropriate methods to measure ischemia may hamper the assessment of the prognosis of nonblanchable erythema (early PI) and make it difficult to distinguish PIs from BE. In the current background, we attempted to examine ischemia in early PI post-decompression and establish a method that enables the early detection of ischemia. For this undertaking, we used hairless rat models and compared the BE and early PI perfusion status by an intravenous injection of Evans Blue (EB) along with a histopathological observation of thrombi. We also used a modified capillary refill time test (CRTT) to detect ischemia in early PI. Our current studies revealed that the presence of ischemia post-decompression accelerated the deterioration of early PI. Importantly, our purpose was to develop a clinically applicable method for the detection of ischemia in early PI.

## 2. Materials and Methods

Animals

Four 12-week-old male hairless rats (HWY/Slc, SLC, Inc., Shizuoka, Japan) were used in this experiment. The animals were given a standard chow diet with ad libitum access to water and in a 12 h light/dark cycle throughout the whole experiment. The protocol was approved by the Committee on the Ethics of Animal Experiments of the Nagano College of Nursing (protocol code: No.2021-5; date of approval: 1 November 2021).

### 2.1. Confirmation, Assessment, and Histological Observation of Ischemia

#### 2.1.1. Generation of BE and Early PI Models

Two rat models representing either blanchable erythema (BE) or early PI were made and analyzed according to the methods of a previous study, with slight modifications [8]. Briefly, hairless rats were anesthetized with isoflurane inhalation (import: 4–5%; maintenance: 2–3%) (Meiji Seika Pharma Co., Ltd. Tokyo, Japan). For the generation of the BE model, the symmetrical part of the left or right back skin was gently lifted and then pressed using two circular neodymium magnets (10 mm in diameter × 4 mm in thickness) (Niroku Seisakusho Co., Ltd., Kobe, Japan) with 440 mmHg of attraction to exert pressure.

The whole process of making these models is shown in Figure 1. In the early PI group, early PI was first modeled on the left dorsal (press for 3 h and 50 min). After early PI decompression for 5 h and 10 min, the BE group was modeled on the right dorsal. After the BE group had been decompressed for 5 min, the following observations and tests were performed simultaneously on both sides (early PI decompression for 6 h). The normal skin before the construction of the rat model was designated as the control.

#### 2.1.2. Macroscopic Observations

Photographs of the wounds were taken with a digital camera (Optio WG-3 PENTAX) post-decompression. The distance between the skin wounds and the camera was set to 20 cm. The optical magnification was set to 4×. The normal skin before the construction of the rat model was designated as the control.

#### 2.1.3. Evaluation of Ischemia

In order to evaluate the degree of ischemia, we injected an EB solution, a biomedical marker used for estimating blood volume [9]. The EB solution (200 mg/kg) was injected via the femoral vein. Observations were made at 0, 5, 10, and 15 min after injection in the erythema in both groups. A spectrophotometer was used to measure the reflectance at 620 nm (based on its maximum absorbance). 

#### 2.1.4. Pathological Analysis

To confirm the presence of thrombi and differentiate between post-mortem clots, after the completion of the EB assessment as described above, we modified the method created by Akyurek [10]. After pulpotomy, with fast chest opening in rats, normal saline (50 mL) was reverse-perfused via the left ventricle (pressure not exceeding 120 mmHg). At the same time, the right atrium was cut. After the liquid turned white, it was filled with 50 mL of ink (111B, Kaimei, Saitama, Japan). The skin tissues were collected and fixed further in a 10% neutral formalin solution. All the specimens were embedded in paraffin and cut into slices (thickness: 5 μm). Sections were stained with eosin staining. Optical microscopy was used to determine whether the blood vessels were filled with red blood cells without ink or thrombi. The number of subcutaneous vessels either in the erythematous area or the control area (approximately 1 cm) and the number of vessels with thrombus were measured via morphometry in both groups. Furthermore, the number of vessels with thrombus was expressed as a percentage of thrombi to the total vessels. Considering that the blood supply and return flow of the rat skin passes through the *panniculus* carnosus muscle [11], the area from the *panniculus* carnosus muscle to 1 mm from the dermis was chosen as the measurement range. 

### 2.2. Measurement of Ischemia—CRTT

#### Capillary Refill Time Test (CRTT)

To perform the CRTT, we started to observe the changes in erythema at the start, 5, 10, 20, 30, and 60 min post-decompression in the BE group. For the early PI model, we observed the changes in erythema at the start, 0.5, 6, 12, 18, 24, 30, 60, and 96 h post-decompression. The right skin of the control group was observed before the generation of the BE model (Figure 2A).

The capillary refill time test (CRTT) is a simple test for examining circulatory insufficiency in patients with sepsis and severe dehydration and also for detecting cutaneous blood perfusion. As a measure of cutaneous perfusion, the importance of the CRTT is based on the fact that skin vascularity is an effective indicator of peripheral vascular status [12]. In our experiment, we used this method to measure blood perfusion. Combined with the continuation of ischemia post-decompression, which can also lead to skin hypoperfusion, we attempted to utilize this method to measure ischemia in early PI [13,14], as described previously. For this purpose, we used a spectrophotometer to record the a* values to objectively assess the changes in erythema during the implementation of the CRTT. The CRTT was implemented using the transparent disc method. According to the method of a previous study [15], the pressure applied in the CRTT in the current study was set at 150 mmHg.

In order to examine simple congestion in BE and ischemia in early PI, we recorded the a* values of two time periods for the observation of perfusion: 1 s and 5 s after depression using the CRTT cut-off values recommended in the relevant literature [12]. The respective perfusion states were reflected by the following ratios (Figure 2B): 1 s after depression/pre-pressure represents the degree of rapid recovery of perfusion; 5 s after depression/pre-pressure represents the degree of delayed perfusion; 1 s after depression/5 s after depression indicates a comparison of the degree of rapid recovery and delay of perfusion.

### 2.3. Statistical Analysis

All data are expressed as the mean ± SEM. Statistical tests were examined using the GraphPad Prism 7.0 software program (GraphPad Software, San Diego, CA, USA). Each dataset was first assessed for normality using the Shapiro–Wilk test. An unpaired-sample *t*-test was used to analyze the data with nonparametric distributions. In all the cases, *p* values of less than 0.05 were considered statistically significant.

## 3. Results

### 3.1. Confirmation of Ischemia after Early PI Decompression

#### 3.1.1. Macroscopic Observation and Spectroscopic Measurement of EB Injection

As shown in Figure 3A, the results showed the coloration of the erythema before and after EB injection. BE was stained blue after EB perfusion post-decompression in comparison to early PI. Erythema could be clearly observed as a reddish in the area of BE and early PI compression before EB injection. Regarding the perfusion at 5, 10, and 15 min after EB injection, the BE group showed rapid perfusion at 5 min, followed by little change in the perfusion volume at 10 and 15 min. In contrast, the early PI group showed slow perfusion at the erythema, with only a small amount of EB staining after 15 min, which reflected a delay in perfusion and a lack of perfusion. These changes were further quantified by measuring the reflectance at 620 nm with a spectrophotometer. The BE group gradually decreased after the injection of EB and reached its lowest reflectance after 5 min. In contrast, the reduction in reflectance was not significant in the early PI group. There was a significant difference between the two groups at different time points (*p* < 0.001).

#### 3.1.2. The Early PI Group Had More Thrombi

To investigate the histological changes in either BE or early PI, the sections of erythema harvested from rats were stained with hematoxylin and eosin. Under a light microscope, we could observe that many small vessels, including both arteries and veins, were filled or congested with erythrocytes. After the injection of ink, these small vessels were either stained with ink deposition or without ink deposition (Figure 4A).

We assumed that ink deposition, representing blood clots, occurred post-mortem, whereas small vessels without ink were considered to be thrombotic. We counted the number of vessels with thrombus and found that the number in the early PI group was significantly greater than that in the BE group.

#### 3.1.3. Observation of Ischemia

To examine the changes in the two groups, we made serial observations using a digital camera. As shown in Figure 5, the erythema of the BE group became apparent at first but gradually faded and almost disappeared at 1 h post-decompression. In contrast, in the early PI group (lower), the erythema persisted until 24 h post-decompression. A slight increase or decrease was observed during the period. At 30 h post-decompression, the erythema began to show punctate erosion and formed a complete ulcer at 96 h.

In the BE group (upper), erythema initially became apparent but gradually faded and almost disappeared at 1 h post-decompression. In contrast, in the early PI group (lower), erythema persisted and developed into an ulcer at 60 h.

### 3.2. Measurement of Ischemia

#### Quantitation of Ischemia by CRTT

Figure 6A shows the a* values measured by a spectrophotometer in each group at certain times after depression. In the normal skin, blood volume refilling was shown to be rapid, and at 5 s of refill time, the volume even exceeded that before pressure loading. The BE group showed rapid refilling, with almost up to maximal blood flow at 1 s after depression and almost no increase at 5 s. The early PI group showed lower refilling blood flow at 1 s after depression and a slight increase at 5 s; however, the amount of refill was still clearly inadequate. This phenomenon was observed throughout the deterioration process, though to varying degrees.

Figure 6B represents the comparison of the a* values ratios of the CRTT in the BE group at 5 min post-decompression (strongest erythema) and 6 h post-decompression (lightest erythema) in the early PI group. Regarding the 5 s/pre-pressure ratio, both the BE and early PI groups exhibited varying degrees of deficient refilling (ratio < 1), compared with the normal skin (ratio > 1). When the 1 s/5 s ratio post-depression was compared, the BE group showed rapid refilling (ratio ≈ 1). In contrast, the early PI group exhibited delayed refilling (ratio < 1). The above results suggest that the early PI group exhibited a greater refilling delay and volume deficit as measured by the CRTT.

## 4. Discussion

The progression of early PI is known to be significantly influenced by ischemia during compression. Even post-decompression, ischemia continues to be involved in the deterioration of PI, although the mechanisms remain to be fully elucidated. In addition, the pallor and low temperature of the skin, which are associated with ischemia, are often overlooked due to the presence of other symptoms (such as redness and fever) and are difficult to identify in the normal clinical setting.

In the current study, we established models of BE and early PI and found that there was insufficient perfusion in early PI post-decompression, as evaluated via an EB injection. In contrast, the BE model showed completely different hyperemia, which differed from early PI. Therefore, we focused on ischemia post-decompression because many pathological changes are intermingled within the lesions post-decompression, including reperfusion injury, secondary inflammation, edema, ischemic hemorrhage, and necrosis. Perhaps, ischemia is one of the most important factors that may be alleviated by aggressive treatment, (e.g., thrombolytic therapy) [5,6]. Many studies have evaluated the presence of ischemia post-decompression and the reasons for its development. Reichel et al. [2] reported that pressure injuries were associated with multiple thromboses. The examinations of pathological specimens showed multiple thromboses in the small vessels of early PI. Moreover, Parish et al. [3] showed that thrombosis was caused by overstretching and injury to the endothelial cells of the blood vessels caused by the deformation of the skin tissue when pressure or shear forces were negatively applied. Gefen et al. [4] reported that ischemia followed by secondary inflammation or thrombosis can be exacerbated by decompression. Experimental animal studies and clinical reports showed that a single-dose ACTH injection can reduce pressure damage by reducing thrombosis [5,6]. Conversely, an increase in the number of vessels with thrombus accelerated tissue damage [16]. On the other hand, Loerakker et al. [17] found that reperfusion after prolonged ischemia may not be complete, thereby continuing the ischemic condition and aggravating tissue damage on MRI. We, therefore, confirmed that the formation of thrombosis is the main cause of ischemia and congestion within early PI lesions. This may provide a pathological basis for the etiology.

It should be pointed out that our results differed from previous findings. For example, the degree of erythema in early PI was rather lighter than in BE [8]. Our previous study found that this may be attributable to the fact that the erythema in early PI is a mixed lesion of multiple circulatory disturbances (e.g., hyperemia, congestion, and hemorrhage) [15]. Therefore, the erythema becomes more severe than the simple congestion of BE. This paradoxical phenomenon suggests that erythema in early PI diminishes through other mechanisms. We presume that ischemia was the main factor, which manifested as the pallor of the skin. The pallor of the skin was masked by more intense erythema and hypothermia, another symptom of ischemia (a symptom of congestion), and this was substituted by increased temperature (a symptom of hyperemia) due to reperfusion and secondary inflammation. Although the MRI method can be used to detect PI, this method is expensive, which makes clinical application difficult. Laser flowmetry is more accurate for assessing blood flow; however, when measuring a mixture of ischemia and congestion with laser flowmetry, the blood flow will show either low or high blood flow. The laser measurement may be too narrow to allow a comprehensive evaluation of early PI. As a result, the lack of a simple and effective method to assess ischemia constitutes a major impediment to our understanding of ischemia, which makes clinical testing difficult and prevents us from further observation and analysis of ischemia.

To this end, we first focused on developing a simple and feasible method for the assessment of ischemia in animal experiments. EB has a long history of use as a biological dye and is widely used in biomedicine because of its properties of high water solubility and slow excretion [9]. EB has been used to determine tumor borders in animal experiments [18] and to evaluate blood volume in vivo [19]. Using this principle, we were able to effectively differentiate between ischemia hypoperfusion and hyperemic hyperperfusion states using EB, providing a method for the identification of early PI and BE (Figure 3). The injection and ink perfusion of EB is not only accurate for the quantitative assessment of ischemia and confirmation of thrombi but also simple and easy to perform. However, the drawback is that its application is currently limited to animal studies and is not used for human detection. In this study, post-mortem coagulation was ruled out with ink perfusion. Ink perfusion solved the problem of the identification of thrombi and post-mortem coagulation.

In addition to the confirmation of ischemia, the measurement of ischemia is also important. The use of the CRTT in the clinical assessment of peripheral microvascular disease and cutaneous microvascular disease has been suggested [20]. However, the CRTT was unable to detect peripheral arterial disease [12]. Furthermore, the CRTT is defined as the time required for a distal capillary bed (fingertip) to regain its original color after receiving firm compression that causes it to blanch and turn white under macroscopic observation [13,14]. Shinozaki et al. [21] performed the CRTT using a pulse oximetry sensor. Not only does this overcome the human factor of macroscopic observation, but it also provides a new method for the quantitative evaluation of the CRTT. We attempted to use a spectrophotometer to record the a* values and implemented the CRTT to evaluate ischemia in early PI and verified that the a* values recorded by the spectrophotometer were a good indicator for assessing the changes in skin blood volume [8].

More importantly, we performed the CRTT with a spectrophotometer and found that the refill delay and deficit persisted throughout the deterioration of the early PI, while the BE group showed rapid refilling. At the same time, we are aware that the complex lesions of hyperemia, congestion, ischemia, and hemorrhage mixed within the lesion can affect the results of the CRTT. The involvement of congestion, for example, leads to a faster refill rate and increased refill volume, while the poor outflow of stasis may lead to a delayed time in the CRTT, as reflected in our results. Therefore, we used a combination of 1 and 5 s of decompression to detect the refill rate and refill volume in the CRTT in order to achieve a comprehensive assessment of the circulatory dynamics in complex circulatory lesions. This is the first report describing the use of the CRTT in the detection of early PI ischemia post-decompression. The CRTT can likely be applied to predict the prognosis of PIs and distinguish PIs from BE. Our previous study [15] used UV light to diagnose hemorrhage in early PI, with significant changes appearing 18 h post-decompression, which predicted deterioration in advance. In contrast, by reliance on the CRTT for the diagnosis of ischemia, which already appears to be significantly different from BE congestion at 0.5 h post-decompression, the time to deterioration in early PI can be predicted earlier.

## 5. Conclusions

This study confirmed that severe perfusion disorder or ischemia caused by thrombi post-decompression is presumed to be the main cause of the deterioration. The CRTT can be used to distinguish early PI from BE. It is expected to be a useful tool for detecting ischemia and predicting the prognosis during the progression of early PI.

Moreover, the CRTT can be performed after the transparent disc method without causing any extra pain or effort and, therefore, is a promising tool for application in clinical settings. However, the histological structures of animal and human skin are very different. Therefore, the methods reported cannot be fully applied to human subjects. There is no “gold standard” testing of early PI. Our proposed CRTT still needs to be validated. In addition, the spectrophotometer that we used, despite its low price, was still a high-cost device, and it is necessary to develop a simpler and lower-cost device for wider application.

## Figures and Tables

**Figure 1 diagnostics-12-02198-f001:**
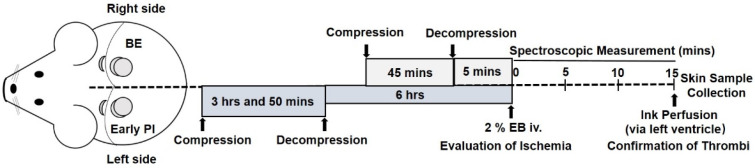
Schematic illustration of blanchable erythema and early pressure injury models in rats. Blanchable erythema (BE) and early pressure injury (PI) were produced in rats by pressing the dorsal skin for different time periods. A 2% Evans Blue (EB) solution was intravenously injected to evaluate ischemia, and ink was perfused from the left ventricle to distinguish thrombi and post-mortem blood clots. i.v.: intravenous injection.

**Figure 2 diagnostics-12-02198-f002:**
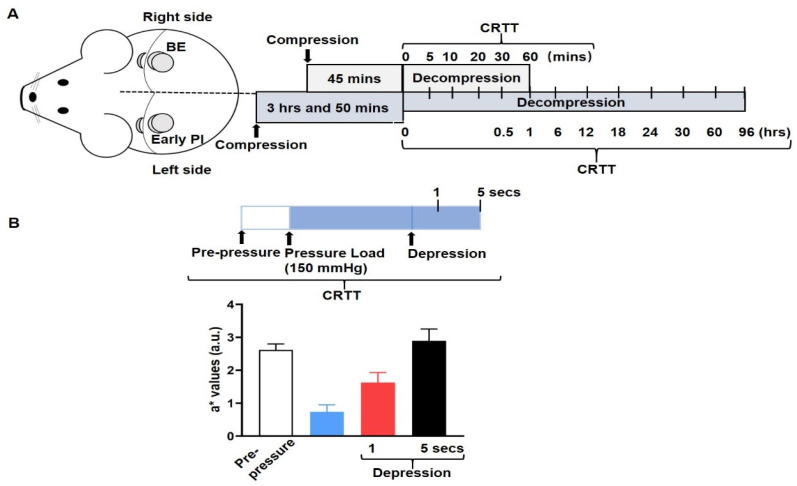
Assessment of ischemia: (**A**) illustration of the procedure for the capillary refill time test (CRTT); (**B**) the CRTT was performed after the transparent disc method by applying temporary pressure (150 mmHg) on the erythema after depression in the BE and early PI group (upper). The a* values represent the blood flow of the following periods: blank indicates pre-pressure, red color indicates 1 s after depression, and black color indicates 5 s post-depression (bottom).

**Figure 3 diagnostics-12-02198-f003:**
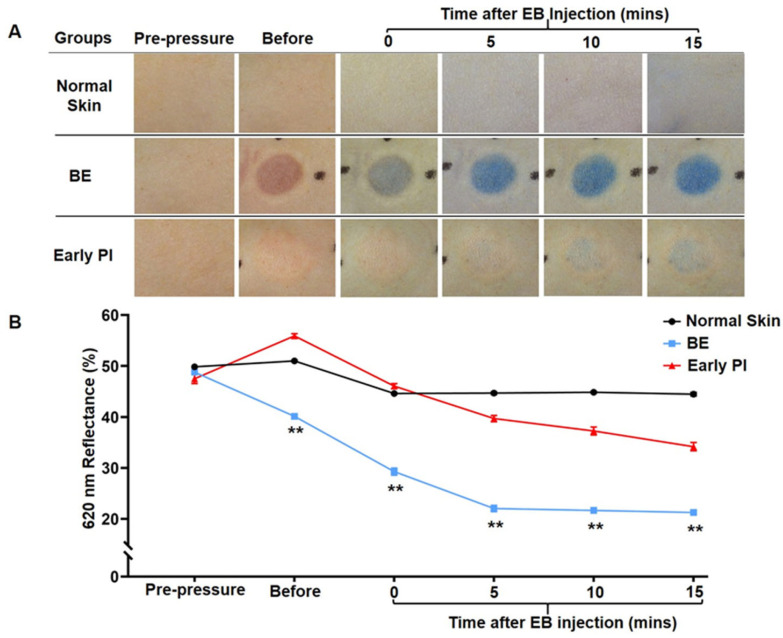
Analysis of BE and early PI after EB injection: (**A**)representative pictures of BE and early PI before and after EB injection. EB solution was injected at 0, 5, 10, and 15 min post-decompression. Normal skin was observed as a normal reference; (**B**) reflectance represented the spectroscopic measurement of BE and early PI at 620 nm absorbance. Data are expressed as the mean ± SEM. n = 8 for each group. ** *p* < 0.001.

**Figure 4 diagnostics-12-02198-f004:**
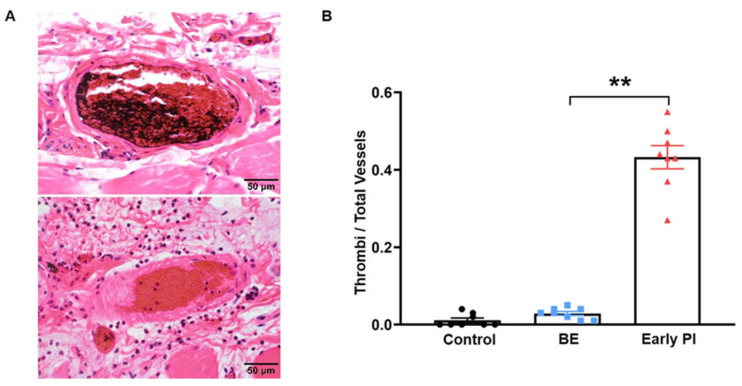
Images of pathological observation and quantification of thrombi: (**A**) the parts with ink perfusion showed post-mortem blood clots. On the other hand, the parts without ink perfusion in the small artery showed thrombus formation post-decompression in the early PI group, which may indicate a cause of circulatory disorder; (**B**) the percentage of thrombus in the three groups. The percentages of arteriovenous thrombus in the total blood vessels of the models were calculated. Values are the mean ± SEM, n = 8 for each group. ** *p* < 0.001.

**Figure 5 diagnostics-12-02198-f005:**
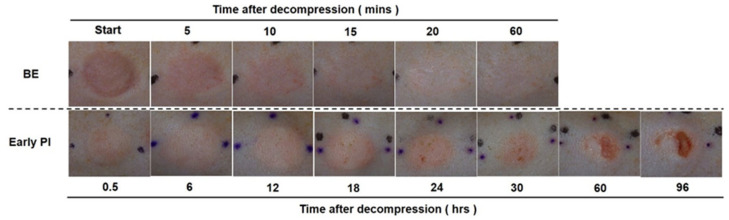
Representative images of macroscopic observation.

**Figure 6 diagnostics-12-02198-f006:**
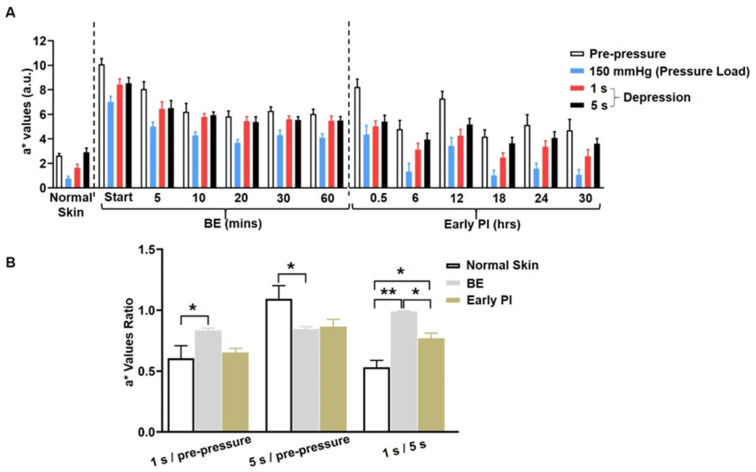
Quantitation of ischemia by the CRTT: (**A**) a* values at certain times post-decompression in the three groups. In this image, 1 s represents the a* values at 1 s after depression, and 5 s represents the a* value at 5 s after depression; (**B**) the ratio of refill time was expressed as above. The a* values were recorded at pre-pressure and 1 s and 5 s after depression. Then, the a* values in the normal skin group, at 0 min post-decompression in the BE group, and 6 h post-decompression in early PI group were chosen. In this image, 1 s after depression/pre-pressure and 5 s after depression/pre-pressure represent the refill time, and 1 s/5 s represents a delayed refill time. * *p* < 0.05, ** *p* < 0.001. n = 8.

## Data Availability

The data that support the findings of this study are available on request from the corresponding author. The data are not publicly available due to privacy or ethical restrictions.

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
