# Peer review of "Elucidation of Ischemic Mechanisms of Early Pressure Injury during Post-Decompression and Detecting Methods"

_diagnostics, 2022, doi:10.3390/diagnostics12092198_

Round 1
Reviewer 1 Report
Line n. 19: twice the word "perfusion".
Line n. 132: it's useful to add more citations about relevant literature.
Line n. 174: change the work "blank" with "black".
Author Response
Thank you for your constructive comments on our manuscript. We have deleted the word “perfusion” and added literature (Ref.12) and corrected “black”.
Reviewer 2 Report
Very Respected Authors,
After carefully reading your manuscript I have few suggestions. Both the abstract and the manuscript are well structurated. The Objective of the manuscript is not sufficiently clear. At the end of the section Introduction, you immediately pointed out what you have developed. I suggest you to write that the objective was to develope a clinically applicable method for the detection of ischemia...
In the section Material and Methods you wrote the number of Decission of the Ethical Committee but not a date. Please, add it.
Author Response
Thank you for your suggestions. We have revised the objective of the manuscript according to your advice (Page2 Line 57).
We have added the approval number of the Ethical Committee (Page2 Line 65).
Reviewer 3 Report
This manuscript is good and technically correct, and presents very detailed aspects, regarding the ischemic mechanisms of early pressure injury during post-decompression. The manuscript is well written and presents useful methods for the study of the aspects analyzed.
This manuscript complements the literature with important data on persistent ischemia, an important cause of early PI deterioration post-decompression, caused by thrombi as it seems in the results.
This manuscript can be accepted for publication after Revision.
In all the text, Justify needs to be used. Consider changing this aspect.
Line 25: in aged societies such … à ageing societies
Line 43: In spite of this, à Despite this… Consider changing the wording.
Line 47: Lack of appropriate methods à The lack…
Line 64: punctuation to end the paragraph.
Line 70: [8].Briefly,… à Consider adding a space before the punctuation!
Line 85: and camera was set à and the camera was se
Line 110: panniculus carnosus muscle à panniculus carnosus muscle (Italic for latin words)
Line 116: In order to à to perform …
Line 269: We therefore confirmed à We, therefore, confirmed
Line 272: from previously findings à from previous findings
Line 289: prevent us from à prevents us from
Line 325: It is likely that the CRTT can à The CRTT can likely…
Hard to read text, please consider rephrasing the sentences.
Line 338: therefore à punctuation ……., therefore,
The introduction needs to be improved.
Figure 1, and 2 needs to be improved for clarity, legends do not respect the template of the journal.
More explanation/justification is needed for the choice of used methods.
Overall, I suggest a minor revision of this work.
Author Response
We appreciate your critiques and expertise. We have the revised manuscript checked by a native and professional staff.
We have revised the Figures 1 and 2 and legends. At the same time, more information about used methods have been added in the method section (Page3 Line 124).